# Antibiotic Resistance Genes in Aerosols: Baseline from Kuwait

**DOI:** 10.3390/ijms24076756

**Published:** 2023-04-04

**Authors:** Nazima Habibi, Saif Uddin, Montaha Behbehani, Mohamed Kishk, Nasreem Abdul Razzack, Farhana Zakir, Anisha Shajan

**Affiliations:** Environment and Life Science Research Centre, Kuwait Institute for Scientific Research, Safat 13109, Kuwait

**Keywords:** antibiotics, antibiotic-resistant bacteria, HT-qPCR, molecular methods, human pathogens

## Abstract

Antimicrobial resistance (AMR) is one of the biggest threats to human health worldwide. The World Health Organization (WHO, Geneva, Switzerland) has launched the “One-Health” approach, which encourages assessment of antibiotic-resistant genes (ARGs) within environments shared by human-animals-plants-microbes to constrain and alleviate the development of AMR. Aerosols as a medium to disseminate ARGs, have received minimal attention. In the present study, we investigated the distribution and abundance of ARGs in indoor and outdoor aerosols collected from an urban location in Kuwait and the interior of three hospitals. The high throughput quantitative polymerase chain reaction (HT-qPCR) approach was used for this purpose. The results demonstrate the presence of aminoglycoside, beta-lactam, fluoroquinolone, tetracycline, macrolide-lincosamide-streptogramin B (MLSB), multidrug-resistant (MDR) and vancomycin-resistant genes in the aerosols. The most dominant drug class was beta-lactam and the genes were *IMP-2-group* (0.85), *Per-2 group* (0.65), *OXA-54* (0.57), *QnrS* (0.50) and *OXA-55* (0.55) in the urban non-clinical settings. The indoor aerosols possessed a richer diversity (Observed, Chao1, Shannon’s and Pielou’s evenness) of ARGs compared to the outdoors. Seasonal variations (autumn vs. winter) in relative abundances and types of ARGs were also recorded (R^2^ of 0.132 at *p* < 0.08). The presence of ARGs was found in both the inhalable (2.1 µm, 1.1 µm, 0.7 µm and < 0.3 µm) and respirable (>9.0 µm, 5.8 µm, 4.7 µm and 3.3 µm) size fractions within hospital aerosols. All the ARGs are of pathogenic bacterial origin and are hosted by pathogenic forms. The findings present baseline data and underpin the need for detailed investigations looking at aerosol as a vehicle for ARG dissemination among human and non-human terrestrial biota.

## 1. Introduction

Antibiotics are crucial medications for treating bacterial infections. With prolonged and indiscriminate usage of antibiotics, antimicrobial resistance (AMR) has developed, which is not only compromising the efficacy of these antibiotics in treating diseases but is also spreading to non-targeted organisms in the environment. Antimicrobial-resistant bacteria (ARBs) and antibiotic-resistance genes (ARGs) have consequently emerged and lingered in the environment [1]. Due to the persistent nature of pharmaceutical residues, the problem is further exacerbated; owing to the mechanism of horizontal gene transfer, the AMR, ARGs and ARBs are spatially reaching regions where no pharmaceutical residues exist [2]. The United Nations Environment Program (UNEP) and World Health Organization (WHO) have listed AMR as one of the major public health risks and a priority environmental issue that needs our immediate attention [3,4].

Antibiotic resistance genes have been reported from differential environmental matrixes such as soil [5,6,7], marine sediments [8,9,10], coastal waters [11,12,13] and river systems [14,15]. Air as a medium for ARG dissemination is less explored [16]. Aerosols are known carriers of many pathogenic microbes [17,18,19,20,21]; it is, therefore, legitimate to speculate that they can be transmitters of ARGs. A study in China has reported the rate of ARG inhalation ranging from 10^3^ to 10^4^ copies per day by an adult [22]. Inhalable ARGs have been reported to occur in urban hospital aerosols and adjacent urban and suburban communities [23] as well as ambient air particulate matter surrounding a municipal solid waste treatment system [24]. In addition, this long- and short-range transport of PM_2.5_ has been reported as a potential AMR distributor [25]. This further strengthens the need to look at ARGs in the inhalable size fraction of air.

It is also prudent to investigate the ARGs in ambient settings [26]. Increased urbanization, population densification, climate change, improper disposal of municipal solid waste and poor sanitization are some factors that are realized as AMR contributors in urban outdoor environments [27,28]. Some examples have indicated aerosols above an estuary [29], soil dust [30], livestock excreta from farms [31,32], wastewater treatment plants [33], hospitals [23,34] and solid waste treatment plants [24] as sources of ARGs in ambient air. Selective pressure in the form of antibiotics, metals and biocides is less expected in ambient air [26]; rather, they are likely to evolve under physicochemical pressures [35] subjected to a higher degree of dispersion and face higher oxidation particularly in the urban atmosphere [36].

The classic methods of ARG detection have been used to look at only a few targeted and cultivable forms since these methods are time and labor-intensive [37]. The culture-independent molecular approach, on the other hand, has emerged as a novel tool to assess ARGs rapidly, precisely and less expensively [38,39,40]. More recently, high throughput quantitative polymerase chain reaction (HT-qPCR) is gaining popularity owing to its capability to identify hundreds of genes in a single run [41,42,43,44]. To underpin the theory of ARG spread through aerosols, we have used a genome-centric approach to assess ARGs associated with size-fractionated aerosols in indoor and ambient air [16]. In the present investigation, we, therefore, employed the HT-qPCR method with the primary objective of identifying clinically relevant ARGs in aerosols collected from an outdoor and indoor location in an urban venue. ARGs were also identified from indoor sites within three hospitals. Spatial and seasonal variations of ARGs were correspondingly studied.

## 2. Results

The qPCR assays positively amplified 52 ARGs in indoor and outdoor aerosols. The Shapiro-Wilk test stipulated our experimental design was balanced with a normal distribution of residuals. Hereafter, we describe the number, types, abundance and spatiotemporal variations of ARGs. Bacterial DNA was found in all the samples. The values from the standard curve predicted bacterial cell counts in the range of 10^4^–10^5^ m^−3^ of aerosols sampled in this study. The month of Oct-21 was exceptional and showed lower bacterial cell counts (10^2^) (Indoor—1.1 × 10^2^ cells m^−3^ air; Outdoor—2.0 × 10^2^ cells m^−3^ air). The mean number of cells m^−3^ air is shown in Figure 1a. Comparisons of average bacterial counts (cells m^−3^ air) between indoor and outdoor sites through ANOVA returned non-significant *p*-values (Tukey’s HSD *p* > 0.05).

### 2.1. Number of ARGs

The number of ARGs in indoor and outdoor samples from KISR was calculated. The numbers were higher in indoor aerosols (mean = 17) compared to outdoor (mean = 12) (Figure 1b). We employed the Kruskal–Wallis H test to compare the mean number of ARGs and recorded a non-significant difference between the two groups, χ^2^(4) = 2.95, *p* > 0.566. Similar observations were recorded through the ANOVA and Tukey’s HSD test (Figure 1b). The temporal variation in ARGs was observed. In indoor aerosols, the highest number of ARGs was recorded in Nov-21 (*n* = 29) > Jan-22 (*n* = 21) > Oct-21 (*n* = 18) > Dec-21 (*n* = 16) > Aug-21 (*n* = 8) > Sep-21 (*n* = 5) and Feb-22 (*n* = 5) (Figure 1c). In outdoor aerosols, the monthly ARG abundances were Sep-21 (*n* = 24) > Dec-21 (*n* = 17) > Oct-21 (*n* = 16) > Feb-22 (*n* = 14) > Nov-21 (*n* = 7) > Jan-22 (*n* = 3) > Aug-21 (*n* = 2) (Figure 1d).

### 2.2. Drug Classes and Gene IDs

The microbial DNA qPCR assay captured genes belonging to the drug classes such as aminoglycoside, beta-lactamases, fluoroquinolones, MLSB, MDR, tetracycline, and vancomycin. The absence of the erythromycin drug class was observed in the samples analyzed. The highest number of genes belonged to the beta-lactamase group both indoors (*n* = 58 ± 3.0) as well as outdoors (*n* = 50 ± 5.1). This class contributed almost 50% of ARGs in the aerosol community. Macrolides (indoor 16 ± 0.7; outdoor 9 ± 0.5) were the second highest in order (Figure 2a). We also looked at the ARG categories present in the corresponding month of sampling and discovered beta-lactamase genes to be present all the time in indoor and outdoor aerosols. It was the only category present in outdoor aerosol in the Aug-21 samples. Two more drug categories, MLSB and fluoroquinolone, were observed in Nov-21 and Jan-22, K-O samples, respectively. In Sep-21, Oct-21, Dec-21 and Feb-22, four or more drug types were found in K-O samples. Similar drug classes were recorded every month in indoor aerosols as well. At least two types were recorded monthly, and six drug classes including aminoglycoside, beta-lactam, fluoroquinolone, MLSB, MDR, and tetracycline, were documented in the Jan-22, K-I sample.

Fifty-two genes were associated with these drug classes and 47 of them were present at least twice. These genes exhibited a correlation ranging between 0.4–1.0 (Pearson’s r^2^; *p* < 0.05) (Figure 2c). We further explored the occurrence of highly prevalent genes in our samples. Five of them exhibited a prevalence > 50% and were considered as part of the core resistome. These were, namely, *IMP-2-group* (0.85), *Per-2- group* (0.65), *OXA-54* (0.57), *QnrS* (0.50) and *OXA-55* (0.55) (Figure 2d). These genes belonging to the beta-lactamase class B (*IMP-2-group*), class A (*Per-2- group*), class D (*OXA 54, OXA 55*), and fluoroquinolones (*QnrS*) that are of known clinical significance and pose a significant risk to human health.

### 2.3. Spatio-Temporal Variations in the Distribution of ARGs

We studied the RA of drug classes and the 47 ARGs in indoor versus outdoor aerosols in two seasons, i.e., autumn (August–November) and winter (December–February). Seasonality had a marked effect on drug classes and ARG abundance, probably because September–October is a transition period with the highest incidences of common flu and allergies in the region [45]. The indoor and outdoor samples in the two seasons had differences in the distribution of ARG classes (Figure 3a). Except for MDR being highly prevalent in the Aug-21, beta-lactams (RA 0.50-1.00) were the pre-dominant class in autumn in both indoor and outdoor samples, whereas their abundance was reduced in winter (RA 0.31–0.81). Aminoglycoside (0.09–0.30) and fluoroquinolones (0.08–0.87) were other gene classes present during winter. Among the ARG distribution in the indoor and outdoor samples, there was significant diversity and variation; almost every sample had a unique ARG profile (Figure 3b). The prevalence and abundance of all 47 ARGs differed between outdoor and indoor aerosols. The hierarchical clustering grouped them into two clusters (Figure 3c).

### 2.4. ARG Richness, Evenness and Ordination Analysis

The alpha diversity indices on observed features (richness) revealed the indices to be higher in indoor aerosols (Figure 4a). The Chao1 index representing the feature evenness inclusive of rare events was also higher in indoor aerosols (Figure 4b). Similarly, the case with Shannon’s diversity index, indicated both richness and evenness greater in indoor aerosols (Figure 4c). However, Pielou’s evenness was comparable in indoor and outdoor aerosols (Figure 4d). Our results suggest the diversity of ARGs was slightly higher in indoor aerosols. This depends on the point source of ARGs in the indoor atmosphere, which is most likely a higher human footfall. Principal coordinate analysis (PCoA) distributed the indoor and outdoor ARG-laden aerosols into overlapping clusters. The F-statistic value was recorded as 0.312 with an R^2^ coefficient of 0.025 at *p* > 0.99. Variations across the first and second axes were 36.8% and 15%, respectively (Figure 4e). The non-significant *p* values obtained during the alpha and beta diversity analysis are attributed to a small sample size. We further employed the DeSeq2 (fitZig model) to look at differentially abundant genes specifically. The zero-inflated Gaussian fit predicted five genes to differ in abundance at a false discovery rate (FDR *q*-value) < 0.05. These genes were *OXA-54*, *QnrD*, *VIM-1 group*, *AAC (6) lb* and *OXA-58 group* (Figure 4f).

A similar analysis was performed, keeping seasonality as an experimental factor. The alpha diversity indices (Observed, Chao1, Shannon and Pielou’s) were consistently higher in the winter season suggesting winter to be favorable for pathogenic forms to proliferate (Figure 5a–d). The PCoA distributed the samples into two partially overlapping clusters (Figure 5e). The F statistics was 1.82, R^2^ of 0.132 at *p* < 0.08 (Axes 1–36.8%; Axes 2–15%). The DeSeq2 identified five genes (*aacC4*, *QnrS*, *QnrD*, *NDM* and *mefA*) significantly abundant at an FDR *(q)* < 0.05. Seasonality was a better driver of variability and diversity among the identified ARGs. These results contrasted with our observations of mean gene numbers, the average number of drug classes and bacterial cell counts. However, it is the type of ARGs and drug classes, more specifically their relative abundances, that vary in space and time rather than their counts.

We compared the alpha diversity indices of indoor aerosols in autumn and winter. The richness (Observed, Chao1, Shannon) and evenness (Shannon, Pielou’s) were higher in the winter season (Figure 6a–d). The PCoA analysis distributed them into a single cluster. The variations among the first and second axis were 43% and 24.6%, respectively (Figure 6e). The R^2^ coefficient was 0.133 at *p* = 0.64. The genes that differed significantly in indoor aerosols between autumn and winter were *oprJ*, *QnrD, VIM-13* and *tetB* at false discovery rate *(q)* < 0.05 (Figure 6f). Likewise, in outdoor aerosols, the alpha diversity (Figure 6g–j) was higher in the winter season. The single cluster in the PCoA plot distributed the samples at 37% at the first axis and 12.6% at the second axis (Figure 6k). The genes that significantly varied (FDR *q* < 0.001) in outdoor aerosols between autumn and winter were *QnrD*, *QnrB-31*, *QnrB-8*, *SFC1* and *QnrS* (Figure 6l). Our results suggested temporal/seasonal variations within the aerosols collected from indoor and outdoor locations.

### 2.5. Evidence of ARGs in Indoor Aerosols of Clinical Settings

We detected the presence of ARGs in indoor aerosols collected from three major hospitals in Kuwait. The volume of air, sampling period and methodical modifications prohibit us from comparing these sites with a non-clinical location. However, hospitals are considered as hot-spots for ARG dissemination and, therefore, it was important to look at the resistomes prevailing at these sites. The drug classes and the corresponding gene identities found at each hospital are mentioned in Table 1. The three hospitals registered different types and numbers of ARGs, with highest (*n* = 27) observed in HI-1 hospital with beta-lactams as the dominant class. Fluoroquinolones were also found in considerable proportions. Aminoglycoside and sulfonamide resistance was completely absent in HI-1 hospital. Interestingly, MDR gene *oprJ* was found at HI-1, raising further concerns about its resistance against multiple drugs. Some other genes, to name a few, were *tetA, vanB*, and *ermB, C*, and *A*, offering resistance to tetracycline, vancomycin and MLSB drug classes, respectively. Although HI-1 and HI-2 hospitals were sampled simultaneously, the numbers were far less (*n* = 15) in the latter. However, HI-2 hospital recorded the presence of a single aminoglycoside gene (*aadA1*) and multidrug resistant *oprJ*. Contrary to HI-1 and HI-2, although a higher volume of air was collected from HI-3 hospital, the numbers of ARGs were less (*n* = 15). It is important to note that samples were collected at different time points and spatial locations; the human footfall at each location is a major contributor to the resistomes of a site. Surprisingly, beta-lactamase, a common aerosol drug class, was completely absent in HI-3 hospital. Chances of primers missing the low copy number genes are quite possible in HT-qPCR assays. Significant numbers of genes showed late amplification with a C_T_ above the set threshold and were, therefore, omitted from the analysis. The source of ARGs in indoor hospital aerosols is most likely the patients and health care staff.

We also looked at the distribution of ARGs in six size fractions. As the Anderson cascade impactor was deployed to collect aerosols from HI-3 hospital, samples of inhalable (2.1 µm, 1.1 µm, 0.7 µm and <0.3 µm) and respirable (>9.0 µm, 5.8 µm, 4.7 µm and 3.3 µm) fractions were analyzed. Intriguingly, ARGs were found in all the size fractions originating from the drug classes of aminoglycoside (0.07–0.40), MLSB (0.09–0.21), quinolones (0.00–0.06), tetracycline (0.35–1.0), MDR (0.20–0.25), sulfonamide (0.00–0.11) and other (0.00–0.11) (Figure 7a). The identified genes were *aadA-1* (0.06–0.15), *cefa_qacelta* (0.00–0.20), *qacE∆1_3* (0.00–0.11), *strB* (0.00–0.31), *sul1_1* (0.00–0.11), *aadA2_3* (0.00–0.20), *ermX_1* (0.09–0.25), *qepA* (0.00–0.06), *sugE* (0.00–0.25) and *tetPA* (0.35–1.0) (Figure 7b). Among the identified genes, *tetPA* was omnipresent in all the size fractions.

## 3. Discussion

The harmful effects of ARGs in aerosols have been realized and initial steps for their monitoring have dawned [26]. The majority of the researchers focus on the characterization of pathogenic microbes, considering them as the only important biological contaminants and often ignore these pernicious pollutants (i.e., ARGs). In this context, the present investigation adds valuable knowledge on the presence and abundance of ARGs in indoor and outdoor aerosols. The HT-qPCR assays proved a technically sound approach to map hundreds of pertinent genes that pose a significant risk to humans [46]. We report their presence in indoor aerosols (*n* = 52) of a research institution’s location in an urban area as well as in ambient air outside the premises. The interiors of three hospitals also had significant numbers of ARGs (*n* = 46). These numbers have also been reported previously in aerosols of urban wastewater treatment plants [22,33].

Antibiotic resistance genes are integrated within the bacterial genomes. Bacterial DNA was found in all the tested samples and the total load was estimated at an average of 10^5^ cells per m^−3^ of air. Slightly lesser bacterial cells (10^4^ m^−3^ air) were recorded in the shower aerosols of a stem cell transplant unit [47] whereas similar cell counts were recorded in eight indoor locations (a classroom, a daycare center, a dining facility, a health center, three houses, an office) in Blacksburg, Virginia [48]. The presence of a comparable number of bacterial cells in outdoor air was in congruence with our previous findings [18,19,49,50]. Similar to our studies, higher numbers of bacteria-like particles have been reported previously in an outdoor location in a university campus in the United States [48]. Temporal seasonal variations in bacterial counts were reported [51], similar to the observations made in this study. In addition to this, the positive amplification of Pan1 and Pan3 bacterial DNA in our samples suggests the presence of bacterial families, Actinobacteria, Bacteroidetes, Euryarchaeota, Firmicutes, Fusobacteria, Proteobacteria, Tenericutes and Spirochaetes. We have previously reported both pathogenic and non-pathogenic bacteria from analogous phyla in aerosol particulate fractions collected from urban, rural and clinical places [17,21,52,53,54,55]. A cocktail of microorganisms was reported in air particulate matter during a smog event in China [56].

The number of ARGs in indoor aerosols (*n* = 97) was 1.2-fold higher than the outdoor aerosols (*n* = 80). In addition, the richness and evenness of ARGs were higher indoors as compared to outdoors (greater alpha diversity parameters). This is expected because the source of ARGs in the indoor air is most probably exhaled breadth [57]. Human footfall is always higher indoors in a country such as Kuwait, due to extreme outdoor weather. Similar to our results, aerosol-associated ARGs were reportedly higher indoors at a university in Tianjin [58]. Some researchers have pointed out that the antibiotic resistome in indoor environments is shaped by the human occupants and not only exhaled breadth. Activities such as skin shedding, feces, saliva and sneezing release ARGs in close vicinity [59,60,61]. In addition, confined animal feeding operations inside a swine, cattle, layer and broiler farm in China exposed the human workers to the aerosolized ARGs in an indoor environment [62]. A couple of studies have also indicated outdoor aerosols as a major contributor to microbes and ARGs indoors [59,61].

The likely sources of ARGs in outdoor aerosols are wave action and wastewater as the area is in close proximity to a marine outfall with hospital waste discharge. This site has maximum concentrations of antibiotics and ARGs in marine sediments and coastal waters [8,9,63]. Increasingly, enriched ARGs were observed along the coastal–urban wastewater treatment plant gradient in ambient air collected in Hong Kong [33]. Submicron aerosols were reported to share antibiotic resistomes (4–17%) with wastewater and sludge in yet another study [64]. KISR is located in close proximity to Sabah Hospital complex and health facilities; therefore, the emanating hospital bioaerosol might be a contributor to ARGs in ambient outdoor aerosols [65].

The most dominant gene classes were beta-lactams, which are often considered the last choice of drug to treat severe bacterial infections such as urinary tract, bloodstream, wound and pneumonia [66,67]. More concerning is the 85% prevalence of *IMP-2 group* (Imipenemase) encoded by the *bla_IMP_* genes and commonly transferred between Gram-negative bacteria through horizontal gene transfer (class 1 or 3 integrons) [68]. A review from a Japanese group revealed about 88 variants of IMP-type metallo-ß-lactamase circulating worldwide, with 25% from Japan, 17% from China and 7% from France [68]. *PER-2-group* [69] and *OXA group* [70] are two other notorious beta-lactams perpetrating globally. These genes are inhabited by the ESKAPEE (*Enterococcus*, *Staphylococcus*, *Klebsiella*, *Acinetobacter*, *Pseudomonas*, *Enterobacter* and *Escherichia*) pathogens on the WHO list for which novel antibiotics are needed [4]. Our ARG-microbial host analysis was in complete agreement with this (Appendix A).

Most of the assumptions regarding health implications are made on PM_2.5_ and below. Presently, we recorded ARGs in a size fraction as low as 0.30 µm both in indoor and outdoor conditions. A study conducted in Harbin; China recorded the presence of ARGs in PM_2.5_ in an urban atmosphere [51]. Further, we discovered ARGs in six size fractions in FW, with higher numbers in the smaller size segment of 0.30 to 0.69 µm. ARGs hosted by ARBs in this size fraction are prone to accumulation in the human airways and might express in the form of illnesses causing hospitalizations. Alternatively, these ARGs carried by the general population in an urban atmosphere pose an elevated risk of disease outbreaks under conducive meteorological conditions [26]. ARGs identified as hospital indoor aerosols pose a significant threat to healthcare professionals, patients and visitors. Hospitals are considered hotspots of resistome transmission [71,72]. Healthcare professionals are at high risk and also act as carriers of these undesirable elements [73,74,75]. Nosocomial outbreaks warrant initiation of continual biomonitoring programs in the region to tackle this serious issue.

We further believe that the stochastic elements comprising organic and inorganic contaminants often interact with inherent microbial communities making the situation even more complex [50,55,76,77,78,79,80,81]. It is also noteworthy to take into consideration the dose metrics and the chemistry of the aerosols in addition to the types and characteristics of ARGs, for a comprehensive risk assessment [26]. Most of the vulnerable microbes and ARGs are present in very low concentrations in aerosols. This study was limited in capturing the mobile genetic elements associated with these ARGs, that play a key role in the dissemination of ARGs via horizontal gene transfer. The alternative approach of next-generation sequencing might overcome the issue for its benefit in capturing the whole genome and the low-abundance genes [82,83].

Seasonal variations in abundance and types of ARGs were observed and were attributed to the differences in bacterial taxonomies. Spatio-temporal variations in microbial profiles of aerosols have been well documented in other regions as well [51,52,84,85]. Ma and group demonstrated variances in bioaerosol-associated ARGs in summer, winter, autumn and spring seasons [51]. We recorded a higher diversity, richness and evenness of ARGs in winter than in autumn in indoor aerosols. This is due to a high peak observed in the number of genes in 21 November coinciding with end of the flu season in Kuwait [45]. Higher pharmaceutical concentrations (including antibiotics) were also recorded in seawater samples collected near storm outlets along the coast of Kuwait. The authors attributed this to increased medical prescriptions during this period [86]. Temperature and humidity are major environmental parameters that fluctuate according to the season, and these are most likely the drivers of resistome variations. To add further, the fluctuations were more pronounced with respect to the types of genes and their relative abundances rather than numbers [87]. Variations in RA of microbial communities in air particulate matter have been reported previously [53].

## 4. Materials and Methods

### 4.1. Sampling

Air samples for the present investigation were collected using a customized device described in detail elsewhere [49,52]. Briefly, a sampler pumped air at the rate of 30 L min^−1^ for 360 min. The sampler consisted of a glass funnel housing a 0.30 µm Anodisc filter (Whatman^®^, Darmstadt, Germany). Aerosols < 0.30 µm were retained on the filter paper and air mass > 0.30 µm was bubbled through three gas washed glass bottles containing sterile phosphate-buffered saline (PBS) (Sigma Aldrich, St. Louis, MO, USA) [52]. The sampler was placed within the Kuwait Institute for Scientific Research (KISR) every week for 360 min from August 2021 until February 2022 covering autumn and winter seasons in Kuwait. The location in KISR was strategically chosen in a corridor that has maximum footfall. Another sampler was co-deployed at an outdoor site in KISR near an AC vent and a coastal walkway. In addition to KISR samples, permission from the Ministry of Health (MOH), Kuwait was granted to collect indoor aerosol samples for a limited duration within three hospital sites (Hospital 1—HI-1, Hospital 2—HI-2 and Hospital 3—HI-3). These hospitals cater to large populations in three major governorates of Kuwait City. The sampling details are provided in Table 2.

### 4.2. Isolation of Nucleic Acids

The indoor and outdoor aerosol in PBS (K-O and K-I) was concentrated by adding 12% beef extract (Sigma, Oslo, Norway) and centrifugation at 10,000× *g* for 30 min. The pellet was treated with 600 µL of lysis buffer (RAV1) at 70 °C for 5 min from the NucleoSpin Mini kit (MachereyNagel, Cedex, France). The lysate was subsequently purified and washed following the manufacturer’s instructions. The Anderson discs collected from Farwaniya Hospital were swabbed several times with sterile PBS to collect the microbial load. The swabs were then processed similarly to the above-mentioned protocol. Aerosols were processed for RNA isolation from HI-1and HI-2 through the standard Trizol^®^ procedure [21]. Nucleic acid quantification was performed on a Qubit fluorometer employing the Qubit HS DNA/RNA assay kits (Thermo Scientific, Waltham, MA, USA). All the RNA samples (HI-1 and HI-2), were converted to complementary DNA (cDNA) before conducting the PCR [20,21]. K-O and K-I samples collected in a month (*n* = 4; one sample per week) were pooled due to their low biological yields.

### 4.3. Quantitative Estimation of Bacterial Cells

Total bacterial counts of K-O and K-I samples were estimated through quantitative polymerase chain reaction (qPCR). Purified DNA (2 μL) was added to 2× iQTM SYBR^®^ Green Supermix (BioRad, Mississauga, ON, Canada) along with universal 16S rRNA bacterial primers (300 nM Forward 5′-CCTACGGGNBGCASCAG-3′; Reverse 5′-GACTACNVGGGTATCTAATCC-3′). The total reaction volume was made up to 20 μL with nuclease-free water and the qPCR was accomplished on the CFX96^TM^ deep well thermal cycler (BioRad, Mississauga, ON, Canada). Standard *Escherichia coli* DNA (2.4 × 10^6^ cells/mL) and nuclease-free water were used as a positive control (PC) and negative controls (NTC), respectively. Serial dilutions (1:10 × 6) of PC were also amplified along with the samples to obtain a standard curve (R^2^ = 0.650; slope = −3.123; y-int = 38.971). The cycling conditions were set for initial denaturation at 95 °C (3 min); followed by 40 rounds of denaturation at 95 °C (20 s); annealing and extension at 60 °C (45 s). A PCR efficiency of 100.0% was achieved. The melt curve analysis was carried out at 60 °C. The cycle threshold (C_T_) values of samples were plotted on the standard curve, to quantify the target gene (16S rRNA gene). This was performed in CFX Maestro^TM^ software (BioRad). The copy numbers of the target gene were then converted to bacterial cells per m^−3^ of air [18].

### 4.4. Microbial qPCR Arrays for ARG Detection

The microbial DNA qPCR assays (Qiagen, Germantown, MD, USA) were used for the detection of 87 ARGs originating from drug classes such as aminoglycoside, beta-lactams (class A, B, C and D), fluoroquinolone/quinolone, erythromycin, macrolide-lincosamide-streptogramin B (MLSB), multidrug resistance (MDR), tetracycline, and vancomycin from the indoor and outdoor aerosols (K-O, K-I, HI-1 and HI-2) collected in the present study. The 25 µL PCR reaction mix comprised 12.5 µL of 1× Master mix, 5 ng of template DNA, and nuclease-free water was loaded on a pre-coated 96-well plate containing the primer and probe sets unique to each ARG [88]. Bacterial DNA (*Escherichia coli*) was included (in triplicate) not only as a reference gene (in wells designated as PPC) but also as a PC to test for the presence of PCR inhibitors and to check the PCR efficiency. In addition, primers and probes associated with PAN bacteria 1 and 3 in the plate served as a second positive control to detect a broad range of bacterial species and confirm the presence of bacterial DNA. A plate with autoclaved nuclease-free water was run as NTC. The cycling conditions were set as initial PCR activation for 10 min at 95 °C followed by 40 cycles of denaturation at 95 °C for 15 s, annealing and extension at 60 °C for 2 min. The PCR was performed on the CFX96^TM^ Deep Well thermal cycler (BioRad). The 2^ΔCT^ method (where ΔC_T_ = C_T_ detected gene − C_T_ 16S rRNA gene) was used to calculate the relative abundances of the detected gene in proportion to the 16S rRNA gene in each sample [44]. Samples exhibiting a C_T_ above 35 and non-specific peaks in the dissociation curve were discarded.

### 4.5. SmartChip^TM^ HT-qPCR Analysis

The DNA isolated from HI-3 hospital (*n* = 2 × 6) was lyophilized and shipped to Resistomap Oy, (Helsinki, Finland) for the HT-qPCR analysis through the SmartChip qPCR assay. The DNA samples were resuspended and 2 ng/µL was added to the PCR reaction mix containing 1 × SmartChip^TM^ green gene expression master mix, 300 nM of primers and nuclease-free water. The primers (*n* = 296) comprised 268 oligos for ARGs (all the above-mentioned classes) and 20 other genes [43]. A total reaction volume of 100 nl was loaded on a SmartChip^TM^ (*n* = 5184 wells) employing an automated SmartChip^TM^ MultiSample NanoDispenser (Takara Bio, San Jose, CA, USA). PCR was conducted on a SmartChip^TM^ Real-Time PCR System (Takara Bio, San Jose, CA, USA). The cycling conditions included initial enzyme activation at 95 °C for 10 min followed by 40 cycles of denaturation at 95 °C for 30 s and annealing at 60 °C for 30 s. A melt curve analysis was performed using the SmartChip^TM^ qPCR software to exclude false positive data (amplicons with unspecific melting curves or multiple peaks). Samples exhibiting a C_T_ above 27 were discarded. RA of amplified genes was calculated by the 2^ΔCT^ method [44].

### 4.6. Statistical Analysis

The Kruskal–Wallis H test was employed to calculate the effect size, test power, outliers and R syntax. One-way ANOVA and Tukey’s HSD were also performed to compare the mean number of ARGs in indoor and outdoor aerosols. The gene list, drug classes and their relative abundances (RA) were exported into the web-based ResistoXplorer [89]. The software’s graphical user interface that implements packages in R (v4.1.0) was used [90]. The ggplot2 (3.3.0), gplots (3.03) igraph (1.2.5), virdis (0.5.1) and RColorBrewer (1.1–2) software packages were used for generating the graphs and box plots. The heat maps were created in the pheatmap (1.2.5) program. The Pearson algorithm (*p* < 0.05) was employed for the correlation pattern search of ARGs [91]. Four alpha diversity parameters (Observed, Chao1, Shannons and Pielou’s indices) accounting for the richness and evenness of genes were estimated [92] and compared by ANOVA. The hierarchical clustering on RA of ARGs was achieved through the WARD algorithm. Ordination analysis (principal coordinate analysis) was performed on Bray Curtis distances between the RA of ARGs applying the permutational analysis of variance (PERMANOVA) [93]. The core resistome analysis was executed on the identified ARGs. The DESeq2 (v1.26.0) was used to predict the differential abundant genes between the experimental parameters [94].

## 5. Conclusions

To summarize the findings of this study, ARGs were found in indoor and outdoor aerosols and exhibited spatial and seasonal variations. Beta-lactams were the most dominant ARG classes that need immediate attention. ARGs were present in hospital aerosols, hence serious thought needs to be given to regular ARG monitoring as humans are at a great risk owing to inhalation of these ARGs. In addition to biomonitoring, stewardship programs to educate the general people, livestock managers and health care consultants are also advisable to minimize the use of antibiotics to tackle this ever-increasing challenge of ARGs and address the One Health Approach proposed by WHO.

## Figures and Tables

**Figure 1 ijms-24-06756-f001:**
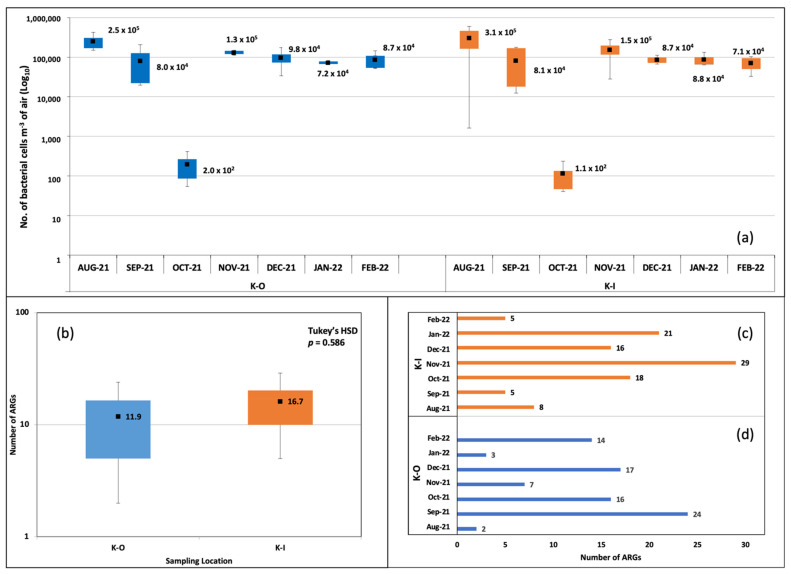
Bacterial cell counts and number of ARGs detected in indoor and outdoor bioaerosols. (**a**) Box plot showing the number of bacterial cells per m^−3^ of air. (**b**) Box plot comparing the mean number of ARGs. (**c**) Number of ARGs in indoor aerosols at the different months of sampling. (**d**) Number of ARGs in outdoor aerosols at the different months of sampling.

**Figure 2 ijms-24-06756-f002:**
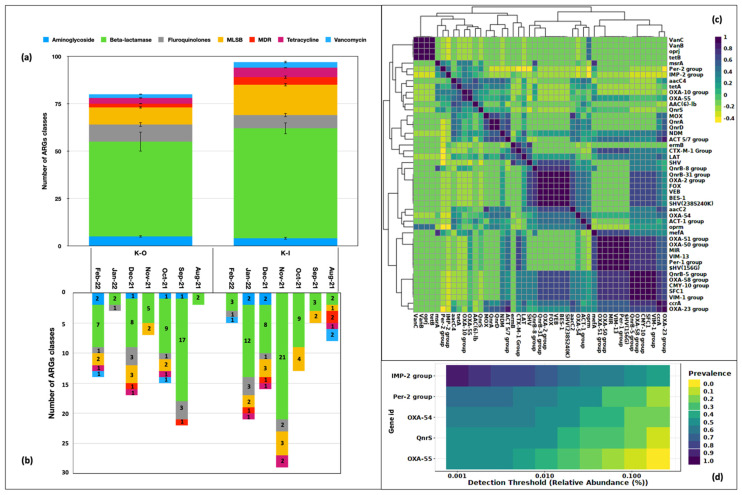
Drug classes identified in bioaerosols. (**a**) Number of ARG of each drug type in indoor and outdoor locations. (**b**) Category of ARGs in the respective month of sampling. (**c**) Types of ARGs. (**d**) Core resistome of bioaerosols.

**Figure 3 ijms-24-06756-f003:**
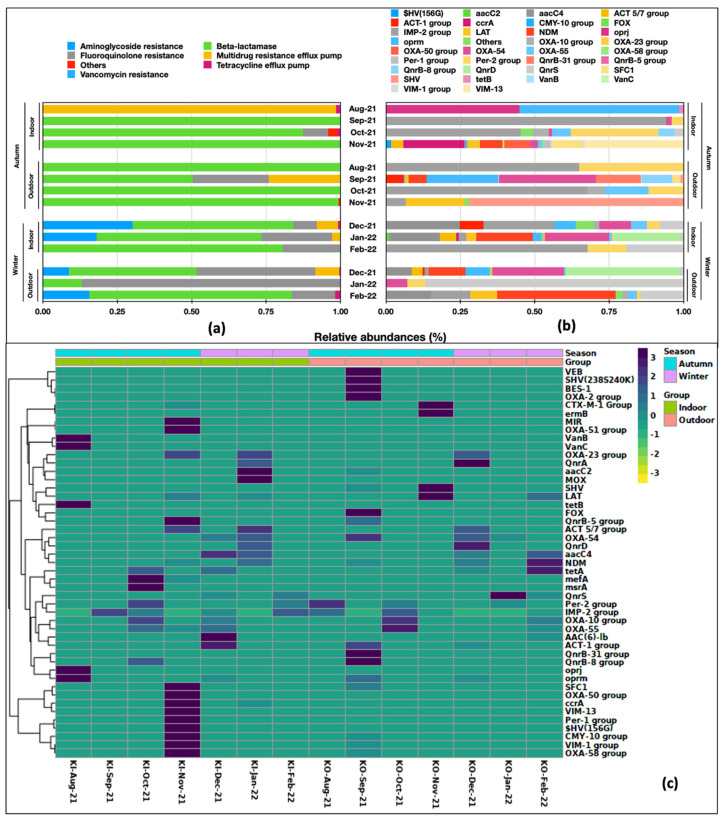
Spatio-temporal variation in relative abundances of: (**a**) drug classes and (**b**) ARGs present in aerosols. (**c**) Hierarchical clustering on Bray Curtis distances.

**Figure 4 ijms-24-06756-f004:**
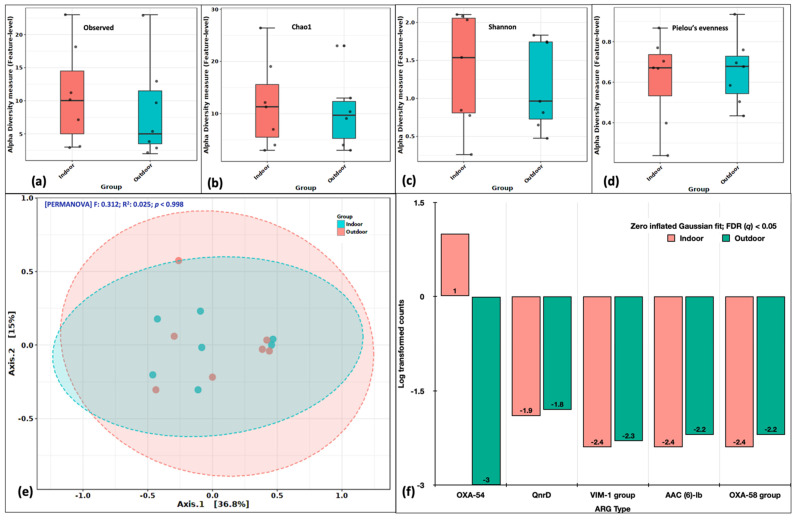
Variations in ARG abundance between indoor and outdoor aerosols. Alpha diversity analysis: (**a**) Observed features, (**b**) Chao1 indices, (**c**) Shannon’s evenness and richness, (**d**) Pielou’s evenness index. Beta diversity analysis: (**e**) Principal coordinate analysis (PCoA) on Bray Curtis distances through Ward algorithm. Differential abundance testing: (**f**) fitZig model (DESeq2). The green colour in each figure represents the indoor samples, whereas the red colour represents outdoor aerosols.

**Figure 5 ijms-24-06756-f005:**
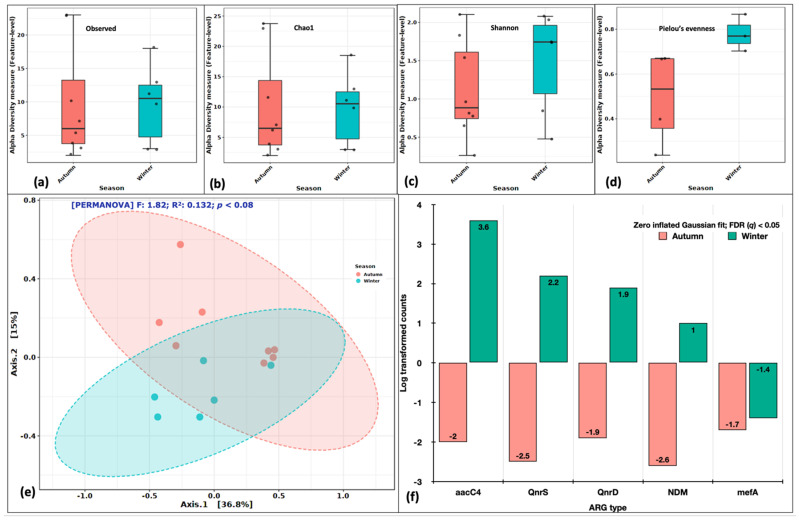
Variations in ARG aerosol abundance between autumn and winter seasons. Alpha diversity analysis: (**a**) Observed features, (**b**) Chao1 indices, (**c)** Shannon’s evenness and richness, (**d**) Pielou’s evenness index. Beta diversity analysis: (**e**) Principal coordinate analysis (PCoA) on Bray Curtis distances through Ward algorithm. Differential abundance testing: (**f**) fitZig model (DESeq2). The green colour in each figure represents the indoor samples, whereas the red colour represents outdoor aerosols.

**Figure 6 ijms-24-06756-f006:**
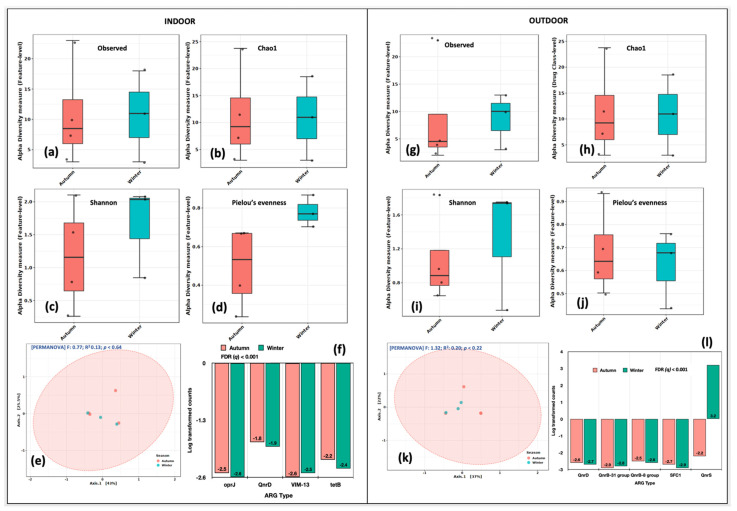
Intra-spatial variations in ARG abundance between autumn and winter seasons. Indoor aerosol alpha diversity analysis: (**a**) Observed features, (**b**) Chao1 indices, (**c**) Shannon’s evenness and richness, (**d**) Pielou’s evenness index. Beta diversity analysis: (**e**) Principal coordinate analysis (PCoA) on Bray Curtis distances through Ward algorithm. Differential abundance testing: (**f**) fitZig model (DESeq2). Outdoor aerosols: (**g**) Observed features, (**h**) Chao1 indices, (**i**) Shannon’s evenness and richness, (**j**) Pielou’s evenness index. Beta diversity analysis: (**k**) Principal coordinate analysis (PCoA) on Bray Curtis distances through Ward algorithm. Differential abundance testing: (**l**) fitZig model (DESeq2). The green colour in each figure represents the indoor samples, whereas the red colour represents outdoor aerosols.

**Figure 7 ijms-24-06756-f007:**
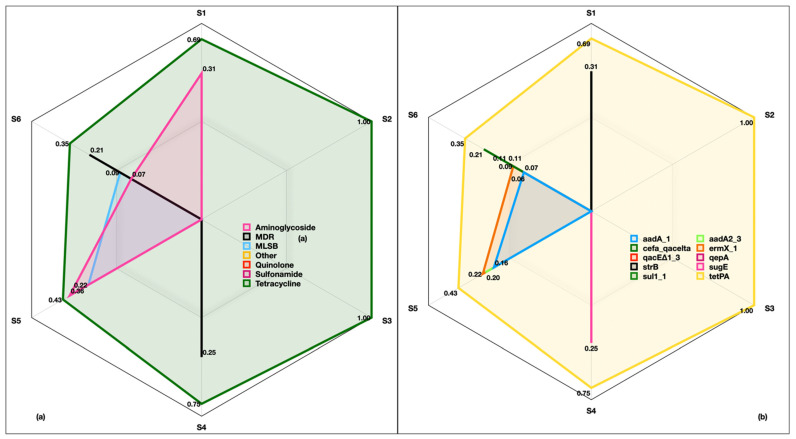
Antibiotic resistance genes in inhalable and respirable size fractions of indoor aerosols collected from HI-3 hospital in Kuwait. (**a**) Relative abundance of drug classes in six size fractions, (**b**) relative abundance of corresponding genes. S1 and S6 represent the six stages of Anderson cascade impactor with a pore size of 0.30 to 0.69 µm (base filter), >0.69 to 1.3 µm (Stage 5), >1.3 to 2.1 µm (Stage 4), >2.1 to 4.2 µm (Stage 3), >4.2 to 10.2 µm (Stage 2) and >10.2 µm (Stage 1).

**Table 1 ijms-24-06756-t001:** ARGs and their drug classes detected in aerosols collected from clinical settings of Kuwait.

Drug Class	Gene ID		
	HI-1	HI-2	HI-3
Aminoglycoside	-	*aadA1*	*aadA; strB; aadA_1; aadA2_3*
Beta-lactams	*CTX-M-9 Group; GES; IMI & NMC-A; SFC1; SHV (156G); SHV (238S240E); IMP-2 group; IMP-5 group; OXA-10 group; OXA-48 group; OXA-55; BES-1; Vim-1 group; OXA-18; OXA-58 group*	*SFC; IMP-5 group; OXA-24 group; OXA-58 group*	-
Fluroquinolones	*QnrB-31 group; QnrB-5 group; QnrB-8 group; QnrC; QnrD*	*QnrC*	*qepA*
MDR	*oprJ*	*oprJ*	*cefa_qacelta; ttgA; sugE*
MLSB	*ermB; ermC; ermA*	*mefA; msrA*	*ermX_1; ermX_2*
Tetracycline	*tetA*	-	*tetPA*
Sulfonamide	-		*sul1_1*
Vancomycin	*VanB*	-	
Other	*-*	-	*qacE∆1_3; ttgB; merA*

Microbial qPCR assay from Qiagen was used to map the genes from HI-1 and HI-2, whereas SmartChip^TM^ HT-qPCR analysis was employed to detect ARGs in six size fractions from HI-3 hospital.

**Table 2 ijms-24-06756-t002:** Sampling locations and types of samples collected for ARG detection.

Location	Sample Type	Sample Code	No. of Samples	Volume of Air (m^3^)	Size Fraction (µm)	Medium of Collection
KISR	Indoor	K-I	7 (pool of 4 samples)	10,800 (43,200)	<0.3	Sterile phosphate-buffered saline
	Outdoor	K-O	7 (pool of 4 samples)	10,800 (43,200)	<0.3	
* Hospital 1	Indoor	HI-1	2 (pool of 2 samples)	3600 (7200)	Whole	Trizol^®^
* Hospital 2	Indoor	HI-2	2 (pool of 2 samples)	3600 (7200)	Whole	
Hospital 3	Indoor	HI-3	12 (*n* = 2 × 6 stages)	10,800	>0.30 to 10.2 μm	Anderson Discs

* Sampling was conducted only for two hours due to the COVID-19 restrictions. Trizol^®^ was used as the collection medium to lyse the live microbial cells, specifically the SARS-CoV-2 virus. Size fractions in the six-stage cascade impactor ranged from 0.30 to 0.69 µm (base filter), >0.69 to 1.3 µm (Stage 5), >1.3 to 2.1 µm (Stage 4), >2.1 to 4.2 µm (Stage 3), >4.2 to 10.2 µm (Stage 2) and >10.2 µm (Stage 1) for samples collected from HI-3 hospital.

## Data Availability

All the data related to this article are presented within the manuscript or provided as Appendix A.

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
