# Peer review of "Antibiotic Resistance Genes in Aerosols: Baseline from Kuwait"

_ijms, 2023, doi:10.3390/ijms24076756_

Round 1
Reviewer 1 Report
In the present manuscript, Habibi Nazima et al. investigated the distribution and abundance of ARGs in indoor and outdoor aerosols collected from an urban location in Kuwait as well as the interior of three hospitals. They have used the high throughput quantitative polymerase chain reaction (HT-qPCR) approach to lookout ARGs in samples collected from these locations. From this study, authors tried to be convinced that ARGs were found in indoor and outdoor aerosols and exhibited spatial and seasonal variations. Beta-lactams were found to be the most dominant ARG classes. Relative abundances and types of ARGs were also recorded in the autumn and winter seasons. Majorly, authors established that there is a serious need for regular ARG monitoring as humans are at great risk owing to the inhalation of these ARGs and also formulate a baseline for further investigation in this direction. Overall, the Study is well-focused and the authors succeeded in documenting the abundance and types of aerosol-generated ARGs. I have some minor concerns and suggestions for authors that could help to improve the manuscript's quality.
Authors have shown the relative Abundance and seasonal variations in ARGs in detail, particularly at KISR. And conducted a similar kind of study at the other three locations (MK, SJ, and FW) in Kuwait but failed to be mentioned whether the trend of abundance and seasonal variations of ARGs at these locations is comparative or any differences they found during this study.
Did the authors compare disease patterns and relative abundances and seasonal variations of ARGs generated through aerosol at KISR? Please discuss it in the discussion section.
The authors have majorly self-cited many references of their previous study in the introduction and discussion section. In my opinion, please try to cite other reference studies wherever appropriate.
Figures 4, 5, and 6 need to align with the text. Part of the figures is missing in the PDF version. Difficult to view figures 4 and 5. Kindly correct the error.
Page 9, Paragraph 2 needs to align. Correct formatting error.
Author Response
Thank you for your positive feedback. A point-by-point response to all the comments have been provided in the attached file

Reviewer 2 Report
The article fits the scope of the journal. It sound scientifically solid and it is quite interesting. There are a few aspects that need to be improved before being published:
a) English grammar needs to be revised, there are a few mistakes here and there. Please revise;
b) The comparison with other similar studies is limited, please broaden your references and touch also studies that applies to other regions or other measurement techniques;
c) The introduction is the worse section. You need to increase it extensively including parts referring to aerosols and aerosol measurement techniques. Examples (and not only) are:
- Baldelli, A., Trivanovic, U., Corbin, J.C., Lobo, P., Gagné, S., Miller, J.W., Kirchen, P. and Rogak, S., 2020. Typical and atypical morphology of non-volatile particles from a diesel and natural gas marine engine. Aerosol and Air Quality Research, 20(4), pp.730-740.
- Sipkens, T.A., Trivanovic, U., Naseri, A., Bello, O.W., Baldelli, A., Kazemimanesh, M., Bertram, A.K., Kostiuk, L., Corbin, J.C., Olfert, J.S. and Rogak, S.N., 2021. Using two-dimensional distributions to inform the mixing state of soot and salt particles produced in gas flares. Journal of Aerosol Science, 158, p.105826.
- Després, V., Huffman, J.A., Burrows, S.M., Hoose, C., Safatov, A., Buryak, G., Fröhlich-Nowoisky, J., Elbert, W., Andreae, M., Pöschl, U. and Jaenicke, R., 2012. Primary biological aerosol particles in the atmosphere: a review. Tellus B: Chemical and Physical Meteorology, 64(1), p.15598.
- Hu, W., Wang, Z., Huang, S., Ren, L., Yue, S., Li, P., Xie, Q., Zhao, W., Wei, L., Ren, H. and Wu, L., 2020. Biological aerosol particles in polluted regions. Current pollution reports, 6, pp.65-89.
- Hybl, J.D., Lithgow, G.A. and Buckley, S.G., 2003. Laser-induced breakdown spectroscopy detection and classification of biological aerosols. Applied Spectroscopy, 57(10), pp.1207-1215.
- Matthias-Maser, S. and Jaenicke, R., 1995. The size distribution of primary biological aerosol particles with radii> 0.2 μm in an urban/rural influenced region. Atmospheric Research, 39(4), pp.279-286.
d) Improve the qualities of the figures.
Author Response
We thank you for your kind review and positive remarks. Please find a point-by-point response to the comments in the attached file.
